# Novel loss functions for ensemble-based medical image classification

**Sivaramakrishnan Rajaraman** \*, Ghada Zamzmi, Sameer K. Antani

National Library of Medicine, National Institutes of Health, Bethesda, MD, United States of America

\* sivaramakrishnan.rajaraman@nih.gov

## Abstract

Medical images commonly exhibit multiple abnormalities. Predicting them requires multi-class classifiers whose training and desired reliable performance can be affected by a combination of factors, such as, dataset size, data source, distribution, and the loss function used to train deep neural networks. Currently, the cross-entropy loss remains the de-facto loss function for training deep learning classifiers. This loss function, however, asserts equal learning from all classes, leading to a bias toward the majority class. Although the choice of the loss function impacts model performance, to the best of our knowledge, we observed that no literature exists that performs a comprehensive analysis and selection of an appropriate loss function toward the classification task under study. In this work, we benchmark various state-of-the-art loss functions, critically analyze model performance, and propose improved loss functions for a multi-class classification task. We select a pediatric chest X-ray (CXR) dataset that includes images with no abnormality (normal), and those exhibiting manifestations consistent with bacterial and viral pneumonia. We construct prediction-level and model-level ensembles to improve classification performance. Our results show that compared to the individual models and the state-of-the-art literature, the weighted averaging of the predictions for top-3 and top-5 model-level ensembles delivered significantly superior classification performance ($p < 0.05$) in terms of MCC (0.9068, 95% confidence interval (0.8839, 0.9297)) metric. Finally, we performed localization studies to interpret model behavior and confirm that the individual models and ensembles learned task-specific features and highlighted disease-specific regions of interest. The code is available at https://github.com/sivaramakrishnan-rajaraman/multiloss_ensemble_models.

## Introduction

Deep learning (DL) has demonstrated superior performance in natural and medical computer vision tasks. Computer-aided diagnostic tools developed with DL models have been widely used in analyzing medical images including Chest-X-rays (CXRs) and computerized tomography (CT). CXRs have been studied extensively where the models are used to predict manifestations of cardiopulmonary diseases such as pneumonia opacities, pneumothorax, cardiomegaly, Tuberculosis (TB), lung nodules, and, more recently, COVID-19 [1, 2]. Such

**Data Availability Statement:** Information on the datasets used in this study can be found within the article in the Materials and methods section, under the heading Datasets. The data are third-party data

and were accessed as described in the article. The authors had no special access privileges.

**Funding:** This study is supported by the Intramural Research Program (IRP) of the National Library of Medicine (NLM) and the National Institutes of Health (NIH). The funder provided support in the form of salaries for authors SR, GZ, and SA, but did not have any additional role in the study design, data collection, and analysis, decision to publish, or preparation of the manuscript. The specific roles of these authors are articulated in the 'author contributions' section.

**Competing interests:** The authors have declared that no competing interests exist.

tools are extremely helpful, particularly in resource-constrained regions where there exists a scarcity of expert radiologists.

The DL model parameters are iteratively modified to minimize the training error using several optimization methods (e.g., stochastic gradient descent). This error is computed using a loss function, also called a cost function, that maps model predictions to their associated costs. Cross-entropy loss is the most commonly used loss function in medical image classification tasks, including CXRs [3–7]. This loss function outputs a class probability value between 0 and 1, where high values indicate high disagreement of the predicted class with the ground truth label. In class-imbalanced medical image classification tasks, training a model to minimize the cross-entropy loss might lead to biased learning since (i) the loss asserts equal weights to all the classes, and (ii) the model would predict the majority of test samples as belonging to the dominant normal class. To mitigate these issues, the authors of [8] proposed a loss function, called focal loss, for object detection tasks. Here, the standard cross-entropy loss function is modified to down-weight the majority background class so the model would focus on learning the minority object samples. Following this study, the focal loss function has been used in several medical image classification studies. For example, the authors of [9] trained DL models to minimize the focal loss and improve pulmonary nodule detection and classification performance using CT scans. They observed that the model trained with the focal loss resulted in superior performance with 97.2% accuracy and 96.0% sensitivity. Another study [10] used the focal loss to train the models toward classifying CXRs into normal, bacterial pneumonia, viral pneumonia, or COVID-19 categories. It was observed that the models trained with the focal loss outperformed other models by demonstrating superior values for precision (78.33%), recall (86.09%), and F-score (81.68%). Aside from these studies, the literature does not have a comprehensive study that investigates the effects of loss functions on medical image classification, particularly CXRs.

DL models learn a mapping function through error backpropagation and update model weights to minimize error. They can vary in their architecture, hyper-parameters, and training strategy, thereby resulting in varying degrees of bias and variance errors. Ensemble learning, a paradigm of machine learning, helps to (i) reduce prediction variance and achieve improved performance over any individual constituent model, and (ii) increase robustness by reducing the range (spread) of the predictions. There are several ensemble methods reported in the literature including majority voting, simple averaging, weighted averaging, and stacking, among others [11]. Ensemble models have been widely used in medical image classification tasks including CXRs [2, 7, 12–16]. However, these studies trained ensemble models to minimize the de-facto cross-entropy loss in their respective classification tasks. To the best of our knowledge, we observed that no studies reported evaluations on the performance of ensemble DL models trained with other loss functions toward improving classification performance.

In this study, we aim to demonstrate the benefits of (i) training DL classification models using existing and proposed loss functions and (ii) constructing model ensembles to improve performance in a multi-class classification task that classifies pediatric CXRs as showing normal lungs, bacterial pneumonia, or viral pneumonia manifestations. This systematic study is performed as follows. First, we train an EfficientNet-B0-based U-Net model on a collection of CXRs and their associated lung masks [17] to segment lungs in the pediatric pneumonia CXR collection [6]. Lung segmentation helps to exclude irrelevant image regions and learn lung region-specific features. We select the EfficientNet-B0-based model because it delivered state-of-the-art (SOTA) performance in ImageNet classification tasks, with reduced computational complexity [18]. Next, the encoder from the trained EfficientNet-B0-based U-Net model is truncated and appended with classification layers. This is done to transfer CXR modality-specific knowledge for improving performance in the task of classifying CXRs in the pediatric

pneumonia CXR dataset into normal, bacterial pneumonia, or viral pneumonia categories. Finally, the top-K (K = 3, 5) performing models are used to construct prediction-level and model-level ensembles. The performance of the individual models, prediction-level, and model-level ensembles are further analyzed for statistical significance. We also performed localization studies to ensure that the individual models and their ensembles learned task-specific features and highlighted the disease-manifested regions of interest (ROIs) in the CXRs.

## Materials and methods

### Datasets

This retrospective study uses the following two datasets:

i. Montgomery TB CXRs [19]: This is a publicly available collection of 58 CXRs showing TB-related manifestations and radiologist readings and 80 CXRs showing lungs with no findings. The images and their associated lung masks are deidentified and exempted from the National Institutes of Health (NIH) IRB review (OHSRP#5357). We use this as an independent test set to evaluate the segmentation model proposed in this study.

ii. Pediatric pneumonia [6]: A set of 4273 CXRs showing lungs infected with bacterial and viral pneumonia and 1583 CXRs showing normal lungs are collected from children of 1 to 5 years of age at the Guangzhou Medical Center in China. The author-defined [6] training set contains 1349, 2538, and 1345 CXRs and the test set contains 234, 242, and 148 CXRs showing normal lungs, bacterial pneumonia, and viral pneumonia manifestations, respectively. The CXRs are acquired as a part of routine clinical care, curated by expert radiologists, and made publicly available with IRB approvals. We use this dataset toward classifying CXRs as showing normal lungs, bacterial pneumonia, or viral pneumonia manifestations.

### Lung segmentation and cropping

As CXR images contain irrelevant regions that do not help in learning classification task-specific features, we segmented the ROI, i.e., the lungs from the CXRs, and used the lung-segmented images for training the classification models. Our review of the literature reveals that U-Net [20] is widely used for segmenting ROIs in natural and medical images. Further, the study of the literature shows that EfficientNet [18] models have achieved superior performance in natural and medical computer vision tasks, as compared to other models, in terms of accuracy, efficiency, and computational complexity. Hence, we used an EfficientNet-B0-based U-Net model [21] to perform pixel-wise segmentation. The EfficientNet-B0-based U-Net model is trained using the CXR collection and their associated lung masks discussed in [17] to minimize the following loss functions: (i) Binary cross-entropy (BCE), (ii) Weighted BCE-Dice [2], (iii) Focal [8], (iv) Tversky [22], and (v) Focal Tversky [23]. We used 10% of the training data for validation with a fixed seed. Each mini-batch of the training data is augmented using random affine transformations such as pixel shifting [-2 +2], horizontal flipping, and rotations [-5 +5] to introduce variability into the training process. The model is trained using an Adam optimizer with an initial learning rate of 1e-3. The learning rate is reduced whenever the validation loss ceased to improve. The model demonstrating the least validation loss is used to predict lung masks of a reduced 512×512 pixel resolution for the CXRs in the Montgomery TB CXR collection. The images are resized using bicubic interpolation from the OpenCV software library. The performance of the segmentation models is evaluated using the following metrics: (i) Segmentation accuracy; (ii) Dice coefficient, and (iii) Intersection over union (IoU). We

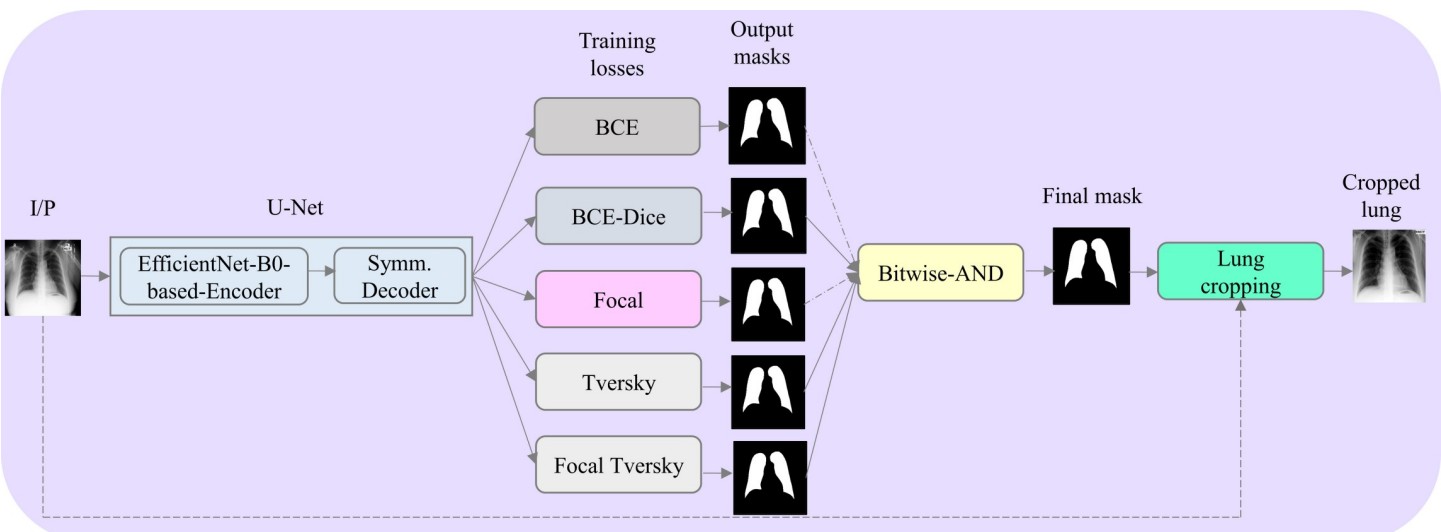

**Fig 1. Segmentation module.** The U-Net constructed with an EfficientNet-B0-based encoder and symmetrical decoder is trained to minimize the following losses: (i) BCE; (ii) Weighted BCE-Dice, (iii) Focal, (iv) Tversky, and (v) Focal Tversky. The trained models predict lung masks in the Montgomery TB CXR collection. The predictions of the top-3 performing models are bitwise-ANDed to produce the final lung mask.

selected the top-3 segmentation models from those that are trained using the aforementioned loss functions based on segmentation accuracy, Dice coefficient, and IoU metrics. The selected models are used to predict the lung masks for the CXRs in the Montgomery CXR collection. These masks are then bitwise-ANDed to produce the final lung mask. The bitwise-AND operation compares each pixel of the predicted masks by the top-3 performing models. If only all the pixels are 1, i.e., belonging to the lung ROI, the corresponding bit in the final mask is set to 1, otherwise, it is set to 0. The final lung mask is then overlaid on the original CXR image to delineate the lung boundaries and the bounding box containing the lung pixels is cropped. The resulting lung-cropped image is resized to 512×512 pixel resolution. Then, the cropped CXRs are contrast-enhanced by saturating the top and bottom 1% of all the image pixels followed by normalizing the pixels to the range [0 1]. Fig 1 shows the diagram of the segmentation module proposed in this study.

## Classification module

The encoder from the trained EfficientNet-B0-based U-Net model is truncated at the 'block5-c_add' layer (TensorFlow Keras naming convention) with feature map dimensions of [16, 16, 512]. This approach is followed to transfer CXR modality-specific knowledge to improve performance in the current CXR classification task. The truncated model is appended with the following layers: (i) a zero-padding (ZP) layer, (ii) a convolutional layer with 512 filters, each of size 3×3, (iii) a global averaging pooling (GAP) layer; and (iv) a final dense layer with three neurons and Softmax activation, to classify the pediatric CXRs as showing normal lungs, bacterial pneumonia, or viral pneumonia manifestations.

We used the train and test splits published in [6] to compare our model performance with the SOTA literature [6, 24]. We allocated 10% of the training data for validation with a fixed seed. The model is trained using a stochastic gradient descent optimizer with an initial learning rate of 1e-3 and momentum of 0.9, to minimize the loss functions discussed in this study. The best-performing model is selected based on the least loss obtained with the validation data. These models are evaluated with the test set, and the performance is recorded in terms of

the following metrics: (a) accuracy; (b) AUROC; (c) area under the precision-recall curve (AUPRC); (d) precision; (e) recall; (f) F-score; and (g) MCC.

The top-K (K = 3, 5) models that deliver superior performance with the test set are used to construct the ensembles. We constructed prediction-level and model-level ensembles. At the prediction level, the models' predictions are combined using various ensemble strategies such as majority voting, simple averaging, weighted averaging, and stacking. In a majority voting ensemble, the most voted predictions are considered final for classifying CXRs to their respective classes. In a simple averaging ensemble, the individual model predictions are averaged to generate the final prediction. For the weighted averaging ensemble, we propose to optimize the weights that minimize the total logarithmic loss so that the predicted labels converge to the target labels. We iteratively minimized the logarithmic loss using the Sequential Least-Squares Programming (SLSQP) algorithm [25]. In a stacking ensemble, the predictions are fed into a meta-learner that consists of a single hidden layer with 9 and 15 neurons respectively, for the top-3 and top-5 performing models. The weights of the top-K models are frozen and only the meta-learner is trained to optimally combine the models' predictions. A dense layer with three neurons and Softmax activation is appended to output prediction probabilities. Fig 2 shows the classification and ensemble frameworks proposed in this study.

For the model level ensemble, the top-K models are instantiated with their trained weights and truncated at their deepest convolutional layer. The features from these layers are concatenated and appended with a 1×1 convolutional layer, to reduce feature dimensions. This is followed by appending a GAP layer and a dense layer with three neurons and Softmax activation to classify the CXRs as showing normal lungs, bacterial pneumonia, or viral pneumonia manifestations. The performance of the individual models, prediction-level ensembles, and model-level ensembles are further compared for statistical significance. All the models are trained and evaluated using Tensorflow Keras 2.4 on a Windows system with an Intel Xeon 3.80 GHz CPU, NVIDIA GeForce GTX 1050 Ti GPU, and CUDA dependencies for GPU acceleration. Statistical significance analysis is performed using R software version 4.1.1.

## Classification losses

We experimented with the following loss functions to provide a comprehensive evaluation of their impact on the multi-class classification task under study: (i) Categorical cross-entropy (CCE) loss; (ii) Categorical focal loss [8]; (iii) Kullback-Leibler (KL) divergence loss [26]; (iv) Categorical Hinge loss [27]; (v) Label-smoothed CCE loss [28]; (vi) Label-smoothed categorical focal loss [28], and (vii) Calibrated CCE loss [29]. We also propose several loss functions, as follows, that mitigate the issues with the existing loss functions when applied to the multi-class classification task under study: (i) CCE loss with entropy-based regularization; (ii) Calibrated negative entropy loss, (iii) Calibrated KL divergence loss; (iv) Calibrated categorical focal loss, and (v) Calibrated categorical Hinge loss. The details of the proposed loss functions are discussed below.

**(i) CCE with entropy-based regularization.** DL models demonstrate low entropy values for the output distributions when they are confident about their predictions [29]. However, under class-imbalanced training conditions, the models might be overconfident about the majority class and classify most of the samples as belonging to this dominant class. This may lead to model overfitting and adversely impact generalization performance. Under these circumstances, a penalty could be introduced in the form of a regularization term that penalizes peaked distributions, thereby reducing overfitting and improving generalization. A model produces a conditional distribution $p_\Omega(y|x)$ through the Softmax function, over a set of classes $y$

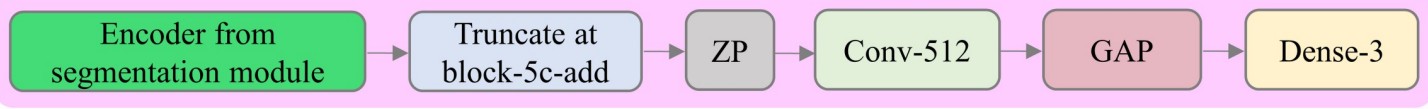

Fig 2. Classification module. The EfficientNet-B0-based encoder is truncated at the block-5c-add layer and appended with the classification layers to output multi-class prediction probabilities. GAP denotes the global average pooling layer and DCL denotes the deepest convolutional layer in the trained models. The classification model is trained to minimize the various loss functions discussed in this study. The top-K (K = 3, 5) performing models are used to construct prediction-level and model-level ensembles.

given an input $x$. The entropy of this conditional distribution is given by,

$$H(p_\Omega(y|x)) = -\sum_k p_\Omega(y_k|x)\log\left(p_\Omega(y_k|x)\right) \tag{1}$$

Here, H denotes the entropy term. A regularization term is proposed where the negative entropy is added to the negative log-likelihood to penalize over-confident output distributions.

It is given by,

$$entropy - reg(\Omega) = -\sum log\ p_{\Omega}(y|x) - \beta H(p_{\Omega}(y|x)) \qquad (2)$$

Here, β controls the intensity of the penalty. Through empirical evaluations, we set the value of β = 2. We used this regularization term in the final dense layer as an activity regularizer and trained the model to minimize the CCE loss.

**(ii) Calibrated negative entropy loss.** We propose an entropy-based loss function where the negative entropy is added as an auxiliary term to the negative log-likelihood term as shown in Eqs [1] and [2] to penalize over-confident output distributions. A model is said to demonstrate poor calibration if it is overconfident or underconfident about its predictions and would not reflect the true occurrence likelihood of the class events. Motivated by [29], we propose to add a regularization term that computes the difference between the accuracy and the predicted probabilities to the entropy-based loss function. This regularization term helps to penalize the model when the entropy-based loss function reduces without a corresponding change in the accuracy. The regularization term forces the accuracy to match the average predicted probabilities, thereby (i) acting as a smoothing parameter that smoothens overconfident or underconfident predictions and (ii) pushing the model to converge to the ideal condition when the accuracy would reflect the true occurrence likelihood. The calibrated negative entropy loss is given by,

$$Calibrated\ negative\ entropy\ loss = -\sum log\ p_{\Omega}(y|x) - \beta H(p_{\Omega}(y|x)) + \lambda.difference \qquad (3)$$

Here, β controls the penalty intensity. The auxiliary term *difference* is calculated for each mini-batch, as given by,

$$difference = |1/K \sum_{k=1}^{K} c_k - 1/K \sum_{k=1}^{K} p(y'_k)| \qquad (4)$$

Here, $y'_k$ denotes the predicted label. The value of $c_k$ is 1 if $y'_k = y_k$; otherwise, $c_k$ is 0. This auxiliary term forces the average value of the predicted probabilities to match the accuracy over all training examples. This pushes the model closer to the ideal situation, where the model accuracy would reflect the true occurrence likelihood of the samples. The auxiliary term serves as a smoothing parameter for predictions with extremely low or high prediction confidences. We tested with different weights for β = [0.00001, 0.0001, 0.001, 0.01, 0.1, 1, 2] and λ = [0.5, 1, 2, 5, 10, 15, 20]. After empirical evaluations, we set the value of β = 0.001 and λ = 10.

**(iii) Calibrated KL divergence loss.** The KL divergence, also called relative entropy, measures the difference between the observed and actual probability distributions. The KL divergence between two distributions $A(x)$ and $B(x)$ is given by,

$$KL\ divergence(A||B) = \sum_{x \in X} A(x) \log\left(\frac{A(x)}{B(x)}\right) \qquad (5)$$

We propose to benefit from the regularization term mentioned in Eq [4] to smoothen model predictions when trained to minimize the KL divergence loss. We propose the calibrated KL divergence loss where the regularization term in Eq [4] is added to the KL divergence loss. This is done to penalize the model when the KL divergence loss reduces without a corresponding change in the accuracy. The calibrated KL divergence loss is given by,

$$Calibrated\ KL\ divergence\ loss = KL\ Divergence(A||B) + \lambda.difference \qquad (6)$$

The auxiliary term *difference* is calculated for each mini-batch and is given by Eq [4]. We tested

with different weights for λ = [0.5, 1, 2, 5, 10, 15, 20]. After empirical evaluations, the value of λ is set to 1.

**(iv) Calibrated categorical focal loss.** The principal limitation of CCE loss is that the loss asserts equal learning from all the classes. This adversely impacts training and classification performance during class-imbalanced training. This holds for medical images, particularly CXRs, where a class imbalance exists between the majority normal class and other minority disease classes. In this regard, the authors of [8] proposed the focal loss for object detection tasks, in which the standard cross-entropy loss function is modified to down weight the majority class so that the model would focus on learning the minority classes. In a multi-class classification setting, the categorical focal loss is given by,

$$Categorical\ Focal\ loss\ (L(k, p)) = -(1 - p'_k)^\gamma \log (p'_k) \tag{7}$$

Here, $K = 3$, denotes the number of classes, $k = \{0, 1, K-1\}$ denotes the class labels for bacterial pneumonia, normal, and viral pneumonia classes respectively, and $p = (p'_0, p'_1, p'_2) \in [0, 1]^3$ is a vector representing an estimated probability distribution over the three classes. The value $\gamma$ denotes the rate at which the easy samples are down-weighted. The categorical focal loss converges to CCE loss at $\gamma = 0$. We propose the calibrated categorical focal loss, where the difference between the accuracy and predicted probabilities is added as a regularization term to penalize the model for overconfident and underconfident predictions when trained to minimize the categorical focal loss. The calibrated categorical focal loss is given by,

$$Calibrated\ categorical\ focal\ loss = -(1 - p'_y)^\gamma \log (p'_y) + \lambda.difference \tag{8}$$

The auxiliary term *difference* is calculated for each mini-batch and is given by Eq [4]. We tested with different weights for $\gamma = [0.5, 1, 2, 5]$ and $\lambda = [0.5, 1, 2, 5, 10, 15, 20]$. After empirical evaluations, the value of $\gamma$ and $\lambda$ is set to 1.

**(v) Calibrated categorical Hinge loss.** The Hinge loss is widely used in binary classification problems to produce "maximum-margin" classification [27], particularly with SVM classifiers. This loss could be used in a multi-class classification setting and is given by,

$$Categorical\ Hinge\ loss = Max(negative - positive + 1, 0) \tag{9}$$

$$negative = Max((1 - y_{true}) * y_{pred}) \tag{10}$$

$$positive = Sum(y_{true} * y_{pred}) \tag{11}$$

Here, $y_{true}$ and $y_{pred}$ denote the ground truth one-hot encoded labels and predictions, respectively. We propose the calibrated categorical Hinge loss, where the difference between the accuracy and predicted probabilities is added as an auxiliary term to the categorical Hinge loss. This auxiliary term penalizes the model when the categorical Hinge loss reduces without a corresponding change in the accuracy. The calibrated categorical Hinge loss is given by,

$$Calibrated\ categorical\ Hinge\ loss = Max(negative - positive + 1, 0) + \lambda.difference \tag{12}$$

The *negative* and *positive* terms are given by Eqs [10] and [11]. The auxiliary term *difference* is calculated for each mini-batch and is given by Eq [4]. We tested with different weights for λ = [0.5, 1, 2, 5, 10, 15, 20]. After empirical evaluations, the value of λ is set to 10.

**Table 1. Segmentation performance achieved by the individual models and the bitwise-ANDed ensemble of the top-3 performing models.**

| Loss/Method | Metrics | | |
|---|---|---|---|
| | IoU | Dice | Accuracy |
| BCE | 0.8186±0.0384 | 0.9571±0.0361 | 0.9720±0.0096 |
| Weighted BCE-Dice | 0.8465±0.0401 | 0.9601±0.0396 | 0.9732±0.0104 |
| Focal | 0.2601±0.0621 | 0.9189±0.0527 | 0.7788±0.0485 |
| Tversky | 0.9360±0.0368 | 0.9624±0.0225 | 0.9912±0.0102 |
| Focal Tversky | 0.9510±0.0415 | 0.9637±0.0271 | 0.9925±0.0130 |
| Ensemble | **0.9518±0.0462** | **0.9652±0.0309** | **0.9927±0.0117** |

The bold numerical values denote the best performance in respective columns.

## Results

### CXR lung segmentation

Recall that an EfficientNet-B0-based U-Net model is trained to minimize BCE, weighted BCE-Dice, focal, Tversky, and focal Tversky loss functions and predict lung masks for the CXRs in the Montgomery TB CXR collection. The lung masks predicted by the top-3 performing models are bitwise-ANDed to produce the final lung mask. The performance of the individual models and the bitwise ANDed model ensemble is evaluated using segmentation accuracy, IoU, and Dice coefficient as shown in Table 1. We observed that the segmentation model demonstrated higher values for the Dice coefficient compared to the IoU metrics due to the way the two functions are defined. The Dice coefficient value is given by twice the area of the intersection of two masks, divided by the sum of the areas of the masks. It is observed from Table 1 that, considering individual models, the segmentation model trained to minimize the focal Tversky loss demonstrated superior performance in terms of IoU, Dice coefficient, and accuracy metrics, followed by those trained with Tversky and weighted BCE-Dice losses. These top-3 performing models are used to construct the ensemble. Here, the lung masks predicted by the top-3 performing models are bitwise-ANDed to produce the final lung mask. We observed that the IoU, Dice coefficient, and accuracy, achieved using the bitwise-ANDed model ensemble are superior compared to any individual constituent model. However, we observed no statistically significant difference in performance ($p > 0.05$) between the individual models and the ensemble.

We used the top-3 performing models and the bitwise-ANDed ensemble approach to predict lung masks for the CXRs in the pediatric pneumonia CXR collection. As the ground truth lung masks for these CXRs are not made available by the authors of [6], the segmentation performance could not be validated. The predicted lung masks are overlaid on the original CXRs to delineate the lung boundaries and are cropped. The cropped images are resized to 512×512 pixel resolution and used for further analysis (i.e., disease classification).

### CXR disease classification

Recall that the encoder from the trained EfficientNet-B0-based U-Net model is truncated and appended with classification layers. This approach is followed to perform a CXR modality-specific knowledge transfer [2, 15, 16, 30] to improve performance in a relevant task of classifying the CXRs in the pediatric pneumonia CXR collection into normal, bacterial pneumonia, or viral pneumonia categories. The classification models are trained to minimize the existing and proposed loss functions in this study. Table 2 summarizes the classification performance achieved by these models. We measured the 95% CI as the exact Clopper–Pearson interval for

**Table 2. Classification performance achieved by the classification models that are trained using the loss functions discussed in this study.**

| Loss | Metrics | | | | | | |
|---|---|---|---|---|---|---|---|
| | Accuracy | AUROC | AUPRC | Precision | Recall | F-Score | MCC |
| CCE | 0.9279 | 0.9921 | 0.9857 | 0.9292 | 0.9279 | 0.9282 | 0.8899 |
| | | | | | | | (0.8653, 0.9145) |
| CCE with entropy-based regularization (β = 2.0) | 0.9311 | 0.9913 | 0.9844 | 0.9337 | 0.9311 | 0.9319 | 0.8953 |
| | | | | | | | (0.8712, 0.9194) |
| KL divergence | 0.9231 | 0.99 | 0.9825 | 0.9261 | 0.9231 | 0.924 | 0.8831 |
| | | | | | | | (0.8578, 0.9084) |
| Categorical focal (γ = 1) | 0.9054 | 0.984 | 0.9753 | 0.9079 | 0.9054 | 0.9054 | 0.8562 |
| | | | | | | | (0.8286, 0.8838) |
| Categorical Hinge | 0.9247 | 0.9892 | 0.9803 | 0.928 | 0.9247 | 0.9255 | 0.8858 |
| | | | | | | | (0.8608, 0.9108) |
| Smoothed-CCE (σ = 0.2) | 0.9231 | 0.9899 | 0.9821 | 0.9252 | 0.9231 | 0.9237 | 0.8829 |
| | | | | | | | (0.8576, 0.9082) |
| Smoothed-focal (σ = 0.2) | 0.9279 | 0.9847 | 0.9744 | 0.9317 | 0.9279 | 0.9287 | 0.8909 |
| | | | | | | | (0.8664, 0.9154) |
| Calibrated-CCE (λ = 10) | **0.9343** | **0.9928** | **0.9869** | **0.9345** | **0.9343** | **0.9338** | **0.8996** |
| | | | | | | | **(0.876, 0.9132)** |
| Calibrated-KL divergence (λ = 1) | 0.9215 | 0.9895 | 0.9817 | 0.9239 | 0.9215 | 0.9217 | 0.8807 |
| | | | | | | | (0.8552, 0.9062) |
| Calibrated focal (γ = λ = 1) | 0.9167 | 0.986 | 0.9777 | 0.9187 | 0.9167 | 0.9164 | 0.8734 |
| | | | | | | | (0.8473, 0.8995) |
| Calibrated Hinge (λ = 10) | 0.9279 | 0.9894 | 0.9803 | 0.9292 | 0.9279 | 0.9275 | 0.8903 |
| | | | | | | | (0.8657, 0.9149) |
| Calibrated negative entropy(β = 1e-3; λ = 10) | 0.9311 | 0.9917 | 0.9851 | 0.9316 | 0.9311 | 0.9308 | 0.8947 |
| | | | | | | | (0.8706, 0.9188) |

The top-K (K = 3, 5) models are selected based on the MCC metric. The values in parentheses denote the 95% CI measured as the exact Clopper–Pearson interval for the MCC metric. Bold numerical values denote superior performance in respective columns.

the MCC metric to test for statistical significance. It is observed that the classification models demonstrated higher values for F-score compared to the MCC metric. F-score provides a balanced measure of precision and recall but could provide a biased estimate since it does not consider TN values. MCC considers TPs, TNs, FPs, and FNs in its computation. The score of MCC lies in the range [-1 +1] where +1 demonstrates a perfect model while -1 demonstrates poor performance. The authors of [31] discuss the benefits of using MCC metric over F-score and accuracy in evaluating classification models. It is observed from Table 2 that the model trained to minimize the calibrated CCE loss demonstrated superior values for accuracy (0.9343), AUROC (0.9928), AUPRC (0.9869), precision (0.9345), recall (0.9343), F-score (0.9338), and MCC (0.8996) metrics. The 95% CI for the MCC metric demonstrated a tighter error margin and hence higher precision as compared to other models. The performance achieved with the calibrated CCE loss is significantly superior ($p < 0.05$) as compared to those achieved by the models that are trained to minimize the categorical focal and calibrated categorical focal loss functions. Fig 3 shows the confusion matrix, AUROC, and AUPRC curves obtained with the calibrated CCE loss-trained model. This performance is followed by the models that are trained to minimize the CCE with entropy-based regularization, calibrated negative entropy, label-smoothed categorical focal, and calibrated categorical Hinge loss functions.

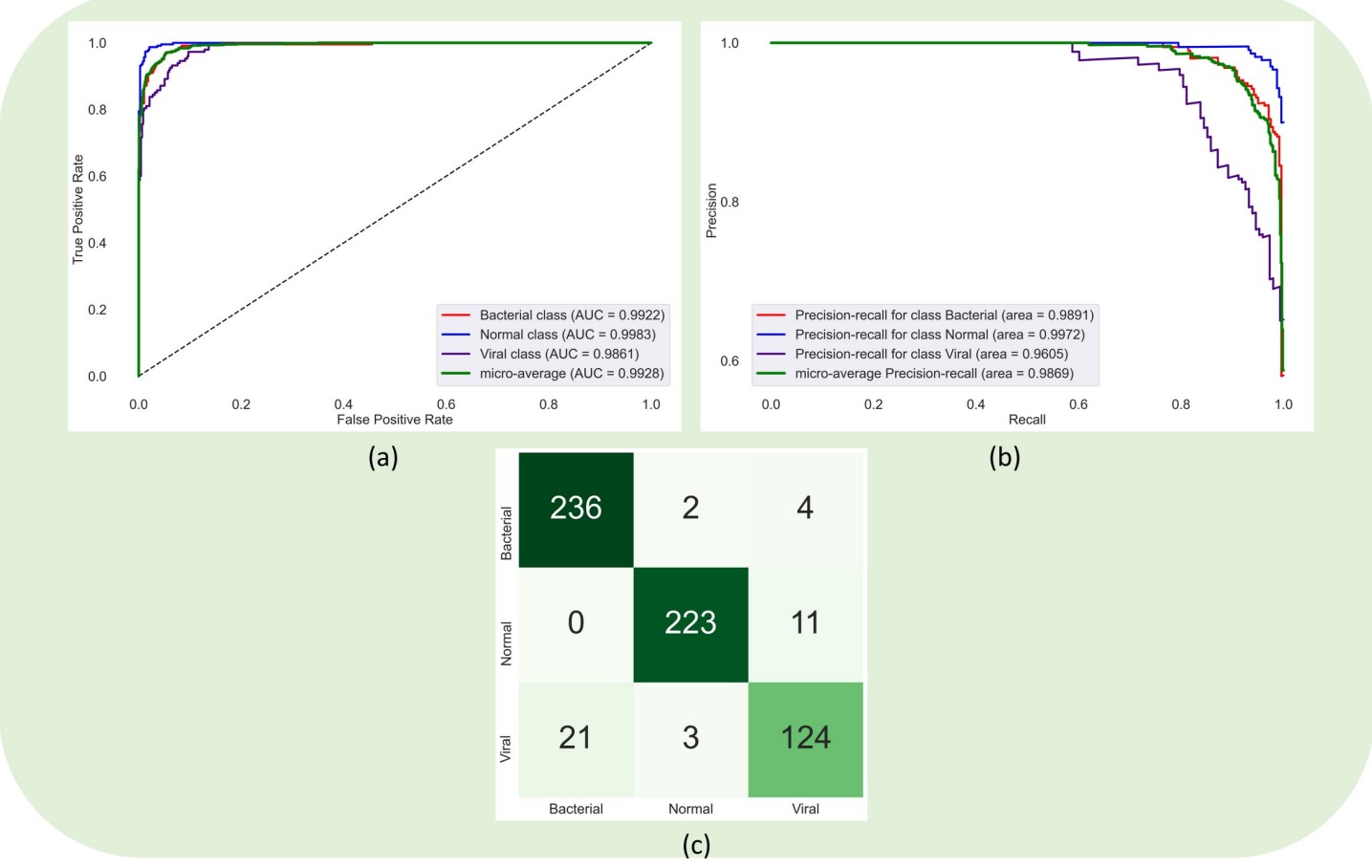

**Fig 3. Confusion matrix, AUROC, and AUPRC curves obtained using the model that is trained to minimize the calibrated CCE loss function.**

The top-3 (i.e., models that are trained to minimize the calibrated CCE, CCE with entropy-based regularization, and calibrated negative entropy losses) and top-5 (i.e., models that are trained to minimize the calibrated CCE, CCE with entropy-based regularization, calibrated negative entropy, label-smoothed categorical focal, and calibrated categorical Hinge losses) are used to construct prediction-level and model-level ensembles. Recall that for the prediction-level ensemble, the models' predictions are combined using majority voting, simple averaging, weighted averaging, and stacking-based ensemble methods. Table 3 summarizes the classification performance achieved by the prediction-level ensembles.

It is observed from Table 3 that the prediction-level ensembles constructed using the top-3 and top-5 performing models demonstrated higher values for F-score as compared to the MCC metrics for the reasons discussed before. The weighted averaging ensemble of the top-5 performing models using the optimal weights [0.40560531, 0.192276399, 0.00356809023, 0.3985502, 1.10927275e-16] calculated using the SLSQP method achieved superior performance compared to other ensembles. The 95% CI obtained using the MCC metric demonstrated a tighter error margin and hence higher precision compared to other ensemble methods. However, we observed no statistically significant difference ($p > 0.05$) in performance across the ensemble methods. Fig 4 shows the confusion matrix, AUROC, and AUPRC curves achieved using the top-5 weighted averaging ensemble.

Recall that the model-level ensembles are constructed using the top-K (K = 3, 5) models by instantiating them with their trained weights and truncating them at their deepest

**Table 3. Performance metrics achieved by the prediction-level ensembles using the top-K (K = 3, 5) models.**

| Models | Method | Metrics | | | | | | |
|--------|--------|---------|---|---|---|---|---|---|
| | | Accuracy | AUROC | AUPRC | Precision | Recall | F-Score | MCC |
| Top-3 | Max voting | 0.9295 | 0.9471 | 0.9412 | 0.9305 | 0.9295 | 0.9297 | 0.8923 |
| | | | | | | | | (0.8679, 0.9167) |
| | Simple averaging | 0.9279 | 0.9924 | 0.9863 | 0.9287 | 0.9279 | 0.9281 | 0.8898 |
| | | | | | | | | (0.8652, 0.9144) |
| | Weighted averaging | 0.9343 | **0.9925** | **0.9865** | 0.9345 | 0.9343 | 0.9338 | 0.8996 |
| | | | | | | | | (0.876, 0.9232) |
| | Stacking | 0.9263 | 0.99 | 0.9831 | 0.9284 | 0.9263 | 0.9269 | 0.8877 |
| | | | | | | | | (0.8629, 0.9125) |
| Top-5 | Max voting | 0.9327 | 0.9495 | 0.9439 | 0.9334 | 0.9327 | 0.9327 | 0.8972 |
| | | | | | | | | (0.8733, 0.9211) |
| | Simple averaging | 0.9295 | 0.9923 | 0.9863 | 0.9311 | 0.9295 | 0.9298 | 0.8926 |
| | | | | | | | | (0.8683, 0.9169) |
| | Weighted averaging | **0.9359** | **0.9925** | **0.9865** | **0.9375** | **0.9359** | **0.9363** | **0.9024** |
| | | | | | | | | **(0.8791, 0.9157)** |
| | Stacking | 0.9279 | 0.9873 | 0.9801 | 0.9303 | 0.9279 | 0.9286 | 0.8903 |
| | | | | | | | | (0.8657, 0.9149) |

The values in parentheses denote the 95% CI measured as the exact Clopper–Pearson interval for the MCC metric. Bold numerical values denote superior performance in respective columns.

convolutional layers. The feature maps from these layers are concatenated and appended with a 1×1 convolutional layer for feature dimensionality reduction. In our study, the feature maps of the deepest convolutional layers for the models have [16, 16, 512] dimensions. Hence, after concatenation, the feature maps for the top-3 models are of [16, 16, 1536] dimensions, and that for the top-5 models are of [16, 16, 2560] dimensions. We used 1×1 convolutions to reduce these dimensions to [16, 16, 512]. The 1×1 convolutional layer is appended with a GAP and dense layer with three neurons to classify the CXRs into normal, bacterial pneumonia, or viral pneumonia categories. Table 4 shows the classification performance achieved in this regard. We observed no statistically significant difference ($p > 0.05$) in performance between the top-3 and top-5 model-level ensembles. We further performed a weighted averaging of the predictions of the top-3 and top-5 model-level ensembles. We calculated the optimal weights [0.3764, 0.6236] using the SLSQP method to improve performance. Fig 5 shows the confusion matrix, AUROC, and AUPRC curves obtained by the weighted averaging ensemble using the predictions of the top-3 and top-5 model-level ensembles. We observed that this ensemble approach demonstrated superior performance for all metrics compared to the individual models and all ensemble methods discussed in this study.

Table 5 shows a comparison of the performance achieved with (i) the weighted averaging ensemble of top-3 and top-5 model-level predictions and (ii) SOTA literature.

The authors of [6] that released the pediatric pneumonia CXR dataset performed binary classification to classify the CXRs as showing normal lungs or other abnormal manifestations. To the best of our knowledge, only the authors of [24] performed a multi-class classification using the train and test splits released by the authors of [6]. We observed that the MCC metric achieved by the weighted averaging ensemble of top-3 and top-5 model-level predictions is significantly superior ($p < 0.05$) compared to the MCC metric reported in the literature [24].

**Disease ROI localization.** We used Grad-CAM tools [32] for localizing the disease-manifested ROIs to ensure that the models learned meaningful features. Fig 6 shows instances of

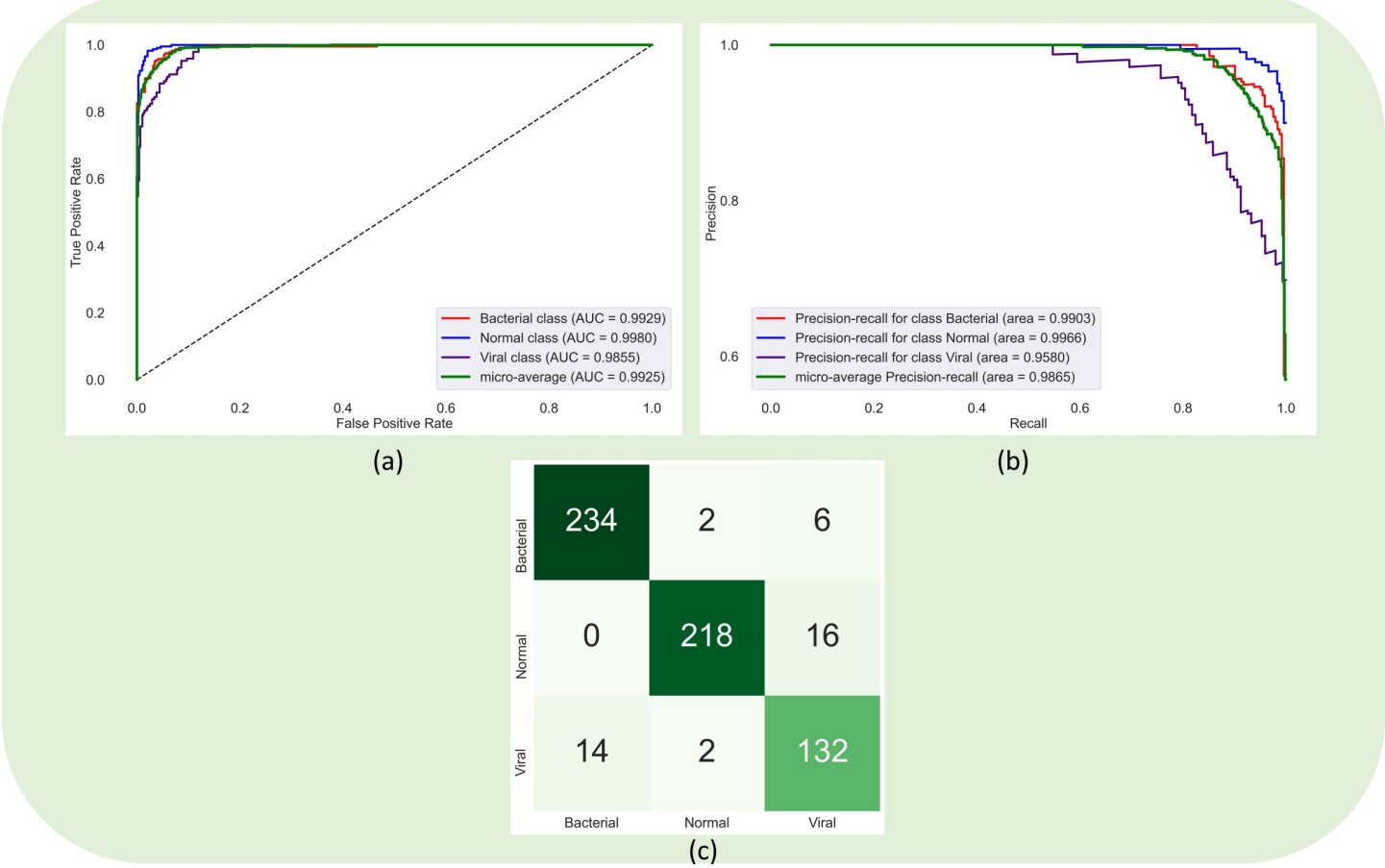

**Fig 4. Confusion matrix, AUROC, and AUPRC curves obtained by the weighted averaging ensemble of the top-5 performing models.**

pediatric CXRs showing expert ground truth annotations for bacterial and viral pneumonia manifestations and Grad-CAM localizations of the top-5 performing models and the top-5 model-level ensemble. It is observed from Fig 6 that the classification models trained using the existing and proposed loss functions and the top-5 model-level ensemble highlighted the ROIs showing disease manifestations. The highest activations, observed as the hottest region in the heatmap, contribute the majority toward the models' decision toward classifying the CXRs into their respective categories.

**Table 4. Classification performance achieved by model-level ensembles.**

| Method | Metrics | | | | | | |
|---|---|---|---|---|---|---|---|
| | Accuracy | AUROC | AUPRC | Precision | Recall | F-Score | MCC |
| Top-3 | 0.9327 | 0.9933 | 0.9881 | 0.9334 | 0.9327 | 0.933 | 0.897 |
| | | | | | | | (0.8731, 0.9209) |
| Top-5 | 0.9359 | 0.9928 | 0.9872 | 0.9365 | 0.9359 | 0.936 | 0.9019 |
| | | | | | | | (0.8785, 0.9253) |
| Weighted averaging | **0.9391** | **0.9933** | **0.9881** | **0.9396** | **0.9391** | **0.9392** | **0.9068** |
| | | | | | | | **(0.8839, 0.9297)** |

The values in parentheses denote the 95% CI measured as the exact Clopper–Pearson interval for the MCC metric.

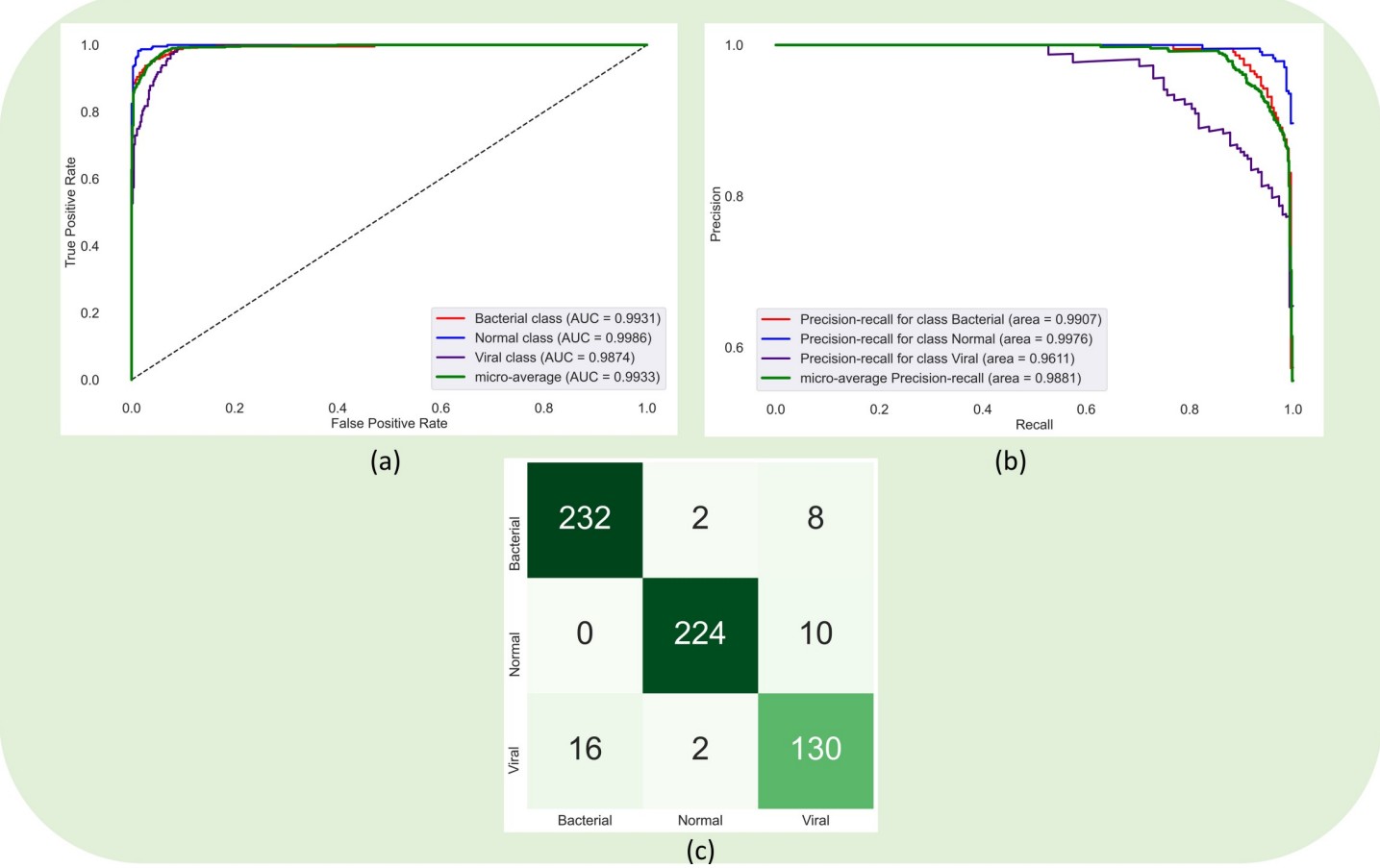

**Fig 5. Confusion matrix, AUROC, and AUPRC curves obtained through the weighted averaging ensemble of the predictions of top-3 and top-5 model level ensembles.**

## Discussion and conclusions

While several studies [33, 34] report using the pediatric pneumonia CXR dataset [6] in a binary classification setting, only the authors of [24] trained models for a multi-class classification task. Further, studies in [33, 34] used ImageNet-pretrained models to transfer knowledge to a target CXR classification task as opposed to a CXR modality-specific pretrained model. Such transfer of knowledge may not be relevant since the characteristics of natural images are

**Table 5. Comparison of the proposed approach with the SOTA literature.**

| Study | Metrics | | | | | | |
|---|---|---|---|---|---|---|---|
| | **Acc.** | **AUROC** | **AUPRC** | **Prec.** | **Rec.** | **F** | **MCC** |
| Kermany et al. [6] | NA | NA | NA | NA | NA | NA | NA |
| Rajaraman et al. [24] | 0.918 | 0.939 | NA | 0.92 | 0.9 | 0.91 | 0.87 |
| | | | | | | | (0.8436, 0.8964) |
| Proposed | **0.9391** | **0.9933** | **0.9881** | **0.9396** | **0.9391** | **0.9392** | **0.9068** |
| | | | | | | | **(0.8839, 0.9297)** |

The values in parentheses denote the 95% CI measured as the exact Clopper–Pearson interval for the MCC metric.

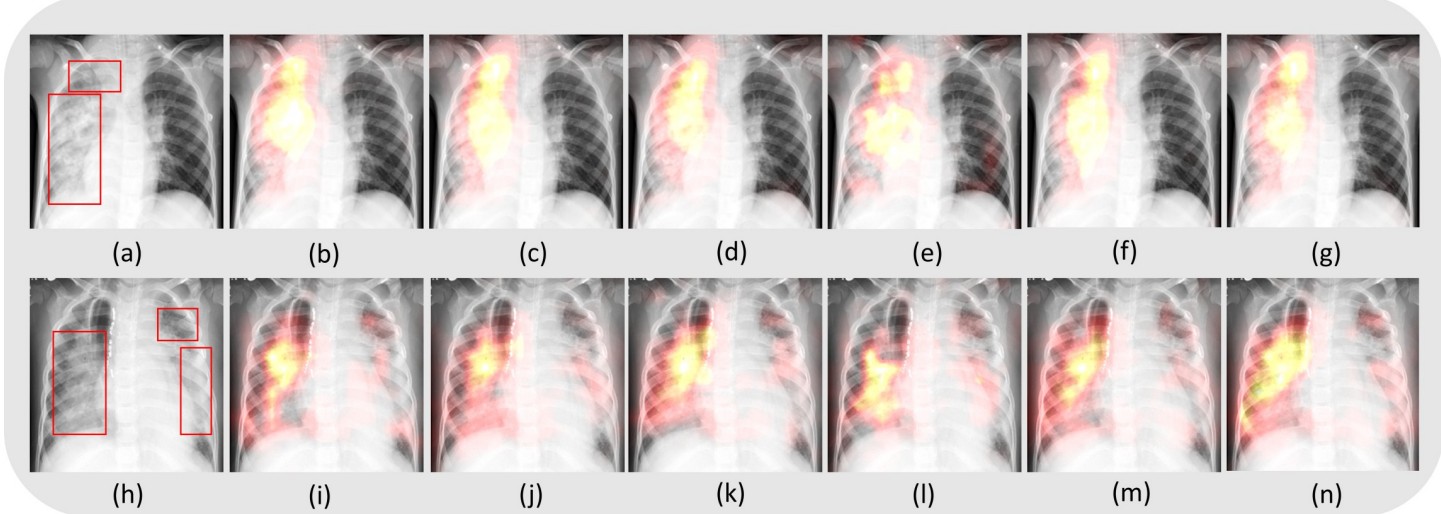

**Fig 6. Grad-CAM-based localization of the disease ROIs.** (a) and (h) denote instances of CXR with expert annotations showing bacterial and viral pneumonia manifestations, respectively. The sub-parts (b), (c), (d), (e), (f), and (g) show Grad-CAM-based ROI localization achieved using the models trained with calibrated CCE, CCE with entropy-based regularization, calibrated negative entropy, label-smoothed categorical focal, calibrated categorical Hinge loss functions, and the top-5 model-level ensemble, respectively, highlighting regions of bacterial pneumonia manifestations. The sub-parts (i), (j), (k), (l), (m), and (n) show the localization achieved using the models in the same order as above, highlighting viral pneumonia manifestations.

distinct from medical images. In this work, we propose to resolve the aforementioned issues by transferring knowledge from a CXR modality-specific pretrained model to improve performance in a relevant CXR classification task. We trained the models using existing loss functions and also proposed several loss functions. Our experimental results showed that the model trained to minimize the calibrated CCE loss demonstrated superior values for all metrics. This performance is followed by those that are trained to minimize the proposed losses such as CCE with entropy-based regularization, calibrated negative entropy, label-smoothed categorical focal, and calibrated categorical Hinge loss.

We evaluated the performance of both prediction-level and model-level ensembles. We observed from the experiments that the model-level ensembles demonstrated markedly improved performance than the prediction-level ensembles. We further improved performance by (i) deriving optimal weights using the SLSQP method, and (ii) using the derived weights to perform weighted averaging of the predictions of top-3 and top-5 model-level ensembles. We observed that the weighted averaging ensemble demonstrated superior performance for all metrics compared to other individual models, their ensemble, and the SOTA literature. Finally, we used Grad-CAM-based visualization tools to interpret the learned weights in the individual models and model-level ensembles. We observed that these models precisely localized the ROIs showing disease manifestations, confirming the expert's knowledge of the problem.

Our study combined the benefits of (i) performing CXR modality-specific knowledge transfer, (ii) proposing loss functions that delivered superior classification performance in a multi-class classification setting, (iii) constructing prediction-level and model-level ensembles to achieve SOTA performance as shown in Table 5. However, there are a few limitations to this study. For example, novel loss functions could be proposed for classification tasks to train models and their ensembles. Other ensemble methods such as blending and snapshot ensembles could also be attempted to improve performance. It is becoming increasingly viable to deploy ensemble models in real-time for image and video analysis with the advent of low-cost

computation, storage solutions, and cloud technology [35]. The methods proposed in this study could be extended to the classification and detection of cardiopulmonary abnormalities [36] including COVID-19, TB, cardiomegaly, and lung nodules, among others.

## Author Contributions

**Conceptualization:** Sivaramakrishnan Rajaraman, Ghada Zamzmi, Sameer K. Antani.

**Data curation:** Sivaramakrishnan Rajaraman.

**Formal analysis:** Sivaramakrishnan Rajaraman, Sameer K. Antani.

**Funding acquisition:** Sameer K. Antani.

**Investigation:** Sameer K. Antani.

**Methodology:** Sivaramakrishnan Rajaraman, Ghada Zamzmi.

**Project administration:** Sameer K. Antani.

**Resources:** Sameer K. Antani.

**Software:** Sivaramakrishnan Rajaraman, Ghada Zamzmi.

**Supervision:** Sameer K. Antani.

**Validation:** Sivaramakrishnan Rajaraman, Sameer K. Antani.

**Visualization:** Sivaramakrishnan Rajaraman.

**Writing – original draft:** Sivaramakrishnan Rajaraman.

**Writing – review & editing:** Sivaramakrishnan Rajaraman, Ghada Zamzmi, Sameer K. Antani.

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
