## [Decision Letter · Decision Letter 0]

8 Nov 2021

PONE-D-21-32872Deep model ensembles with novel loss functions for multi-class medical image classificationPLOS ONE

Dear Dr. Rajaraman,

Thank you for submitting your manuscript to PLOS ONE. After careful consideration, we feel that it has merit but does not fully meet PLOS ONE’s publication criteria as it currently stands. Therefore, we invite you to submit a revised version of the manuscript that addresses the points raised during the review process.

ACADEMIC EDITOR: Based on the comments from the reviewers and my own observation I recommend major revisions.

We look forward to receiving your revised manuscript.

Kind regards,

Thippa Reddy Gadekallu

Academic Editor

PLOS ONE

Journal Requirements:

"This study is supported by the Intramural Research Program (IRP) of the National

Library of Medicine (NLM) and the National Institutes of Health (NIH). The intramural research scientists (authors) at the NIH dictated study design, data collection, data analysis, decision to publish and preparation of the manuscript."

We note that one or more of the authors is affiliated with the funding organization, indicating the funder may have had some role in the design, data collection, analysis or preparation of your manuscript for publication; in other words, the funder played an indirect role through the participation of the co-authors. If the funding organization did not play a role in the study design, data collection and analysis, decision to publish, or preparation of the manuscript and only provided financial support in the form of authors' salaries and/or research materials, please do the following:

a. Review your statements relating to the author contributions, and ensure you have specifically and accurately indicated the role(s) that these authors had in your study. These amendments should be made in the online form.

b. Confirm in your cover letter that you agree with the following statement, and we will change the online submission form on your behalf: 

“The funder provided support in the form of salaries for authors SR, GZ and SA, but did not have any additional role in the study design, data collection and analysis, decision to publish, or preparation of the manuscript. The specific roles of these authors are articulated in the ‘author contributions’ section.

"This study is supported by the Intramural Research Program (IRP) of the National Library of Medicine (NLM) and the National Institutes of Health (NIH)."

"This study is supported by the Intramural Research Program (IRP) of the National

Library of Medicine (NLM) and the National Institutes of Health (NIH). The intramural research scientists (authors) at the NIH dictated study design, data collection, data analysis, decision to publish and preparation of the manuscript."

Additional Editor Comments:

The authors are suggested to address all the comments carefully. The authors can cite the references suggested by the reviewers only if they are relevant and strengthen the references section.

Reviewers' comments:

Reviewer's Responses to Questions

**Comments to the Author**

1. Is the manuscript technically sound, and do the data support the conclusions?

Reviewer #1: Yes

Reviewer #2: No

Reviewer #3: Partly

2. Has the statistical analysis been performed appropriately and rigorously? 

Reviewer #1: No

Reviewer #2: I Don't Know

Reviewer #3: Yes

3. Have the authors made all data underlying the findings in their manuscript fully available?

Reviewer #1: No

Reviewer #2: No

Reviewer #3: Yes

4. Is the manuscript presented in an intelligible fashion and written in standard English?

Reviewer #1: Yes

Reviewer #2: No

Reviewer #3: Yes

5. Review Comments to the Author

Reviewer #1: The proposed work presents a deep model ensembling approach with loss functions for multiclass image classification. There have been numerous research works is already conducted in this domain. However, the approach brings some interesting discussions about the application of loss functions in medical images. However, the following revisions are required.

• The title of the paper needs revision. The length of the paper is very long, and I recommend authors to focus on the essential parts and discard the basic stuff such as what is ensemble deep learning, what is statistical analysis, etc.

• Secondly, the manuscript should be focused on the loss functions for multiclass image classification. But the paper discusses so many other things, such as disease classification details. If authors want to keep these contents, then organize the manuscript so that it is easy for readers to read. I recommend, authors to revise the entire manuscript with the focus on “with loss functions for multiclass image classification.”

• The literature review carried out for the proposed work is not up to date. The proposed research demands the referral of some of the latest research works published recently, such as “ReCognizing SUspect and PredictiNg ThE SpRead of Contagion Based on Mobile Phone LoCation DaTa (COUNTERACT): A System of identifying COVID-19 infectious and hazardous sites, detecting disease outbreaks based on the internet of things, edge computing, and artificial intelligence”, “Histogram of Oriented Gradient-Based Fusion of Features for Human Action Recognition in Action Video Sequences

Reviewer #2: The work lacks novelty. Literature review is poor. Proposed work should be described clearly with clear diagram/algorithm along with discussion.Introduction section and conclusion need to be revised.

I strongly suggest the authors to format the content and structure of the paper before submission. This article does not look like a research paper, just like a manual. Simultaneously, the figures included in this paper are not obvious and casual. The authors should refer to some related papers in some venue for revision.

Reviewer #3: Abstract should be concise yet. But should give complete overview of the work and study.

Abstract should reflect the background knowledge on the problem addressed need to be added.

Abstract should reflect the wide range of applications and its possible solutions need to be added.

In Introduction section, the drawbacks of each conventional technique should be described clearly.

Introduction section can be extended to add the issues in the context of the existing work

What is the motivation of the proposed work?

Literature review techniques have to be strengthened by including the issues in the current system and how the author proposes to overcome the same

Research gaps, objectives of the proposed work should be clearly justified.

The writing of the paper needs a lot of improvement in terms of grammar, spellings, and presentations. The paper needs careful English polishing since there are many typos and poorly written sentences.

Authors can use latest related works from reputed journals like IEEE/ACM Transactions, MDPI, Elsevier, Inderscience, Springer, Taylor & Francis etc. and write the references in proper format, from year 2020-21.

The authors seem to disregard or neglect some important finding in results that have been achieved in paper. So, elaborate and explain the results in more details.

Improve the results and discussion section in paragraph.

The conclusion should state scope for future work.

6. PLOS authors have the option to publish the peer review history of their article (what does this mean?). If published, this will include your full peer review and any attached files.

Reviewer #1: No

Reviewer #2: No

Reviewer #3: No

---

## [Author Response · Author response to Decision Letter 0]

19 Nov 2021

Response to the Editor:

We render our sincere thanks to the Editor for arranging peer review and encouraging resubmission of our manuscript. To the best of our knowledge and belief, we have addressed the concerns of the Editor and the reviewers in the revised manuscript.

Q1: Please ensure that your manuscript meets PLOS ONE's style requirements, including those for file naming. 

Author response: We have formatted the manuscript per the templates recommended by the Editor. 

Q2: We note that the grant information you provided in the ‘Funding Information’ and ‘Financial Disclosure’ sections do not match. When you resubmit, please ensure that you provide the correct grant numbers for the awards you received for your study in the ‘Funding Information’ section. Thank you for stating the following financial disclosure: "This study is supported by the Intramural Research Program (IRP) of the National Library of Medicine (NLM) and the National Institutes of Health (NIH). The intramural research scientists (authors) at the NIH dictated study design, data collection, data analysis, decision to publish, and preparation of the manuscript." We note that one or more of the authors is affiliated with the funding organization, indicating the funder may have had some role in the design, data collection, analysis, or preparation of your manuscript for publication; in other words, the funder played an indirect role through the participation of the co-authors. If the funding organization did not play a role in the study design, data collection, and analysis, decision to publish, or preparation of the manuscript and only provided financial support in the form of authors' salaries and/or research materials, please do the following: a. Review your statements relating to the author contributions and ensure you have specifically and accurately indicated the role(s) that these authors had in your study. These amendments should be made in the online form. b. Confirm in your cover letter that you agree with the following statement, and we will change the online submission form on your behalf: “The funder provided support in the form of salaries for authors SR, GZ, and SA, but did not have any additional role in the study design, data collection, and analysis, decision to publish, or preparation of the manuscript. The specific roles of these authors are articulated in the ‘author contributions’ section.

Author response: All authors of this manuscript are employed by the National Library of Medicine. Our research is supported by the Intramural Research Program (IRP) of the National Library of Medicine (NLM) and the National Institutes of Health (NIH). We do not have a specific grant number. All authors reviewed the contributions listed in the manuscript. We hereby agree to include the following statements under the “Funding Information and Financial Disclosure” sections in the online submission form.

 “This study is supported by the Intramural Research Program (IRP) of the National Library of Medicine (NLM) and the National Institutes of Health (NIH). The funder provided support in the form of salaries for authors SR, GZ, and SA, but did not have any additional role in the study design, data collection, and analysis, decision to publish, or preparation of the manuscript. The specific roles of these authors are articulated in the ‘author contributions’ section.”

Q3: Thank you for stating the following in the Acknowledgments Section of your manuscript: "This study is supported by the Intramural Research Program (IRP) of the National Library of Medicine (NLM) and the National Institutes of Health (NIH).” We note that you have provided funding information that is not currently declared in your Funding Statement. However, funding information should not appear in the Acknowledgments section or other areas of your manuscript. We will only publish funding information present in the Funding Statement section of the online submission form. Please remove any funding-related text from the manuscript and let us know how you would like to update your Funding Statement

Author response: We have removed the Acknowledgment section (and included text) per the Editor’s recommendation. 

Response to Reviewer #1:

We render our sincere thanks to the reviewer for the valuable comments and appreciation of our study. To the best of our knowledge and belief, we have addressed the reviewer’s concerns. 

Q1: Is the manuscript technically sound, and do the data support the conclusions? Yes; Has the statistical analysis been performed appropriately and rigorously? No; Have the authors made all data underlying the findings in their manuscript fully available? No; Is the manuscript presented in an intelligible fashion and written in standard English? Yes.

Author response: 

We wish to reiterate, as indicated in the manuscript, that the data used in this study is publicly available without restriction. The details of the data and their availability are discussed under the Materials and methods section. We compared models’ performance and reported statistical significance in the results. We computed the binomial confidence intervals as the exact Clopper–Pearson interval for the MCC metric to analyze statistical significance. These results are comprehensively discussed in the revised manuscript. 

Q2: The proposed work presents a deep model ensembling approach with loss functions for multiclass image classification. There have been numerous research works is already conducted in this domain. However, the approach brings some interesting discussions about the application of loss functions in medical images. 

Author response: We sincerely thank the reviewer for the words of appreciation. The study aims to compare multi-class classification performance using the models trained on existing and novel loss functions proposed in this study. We propose several loss functions including cross-entropy loss, negative entropy loss, KL divergence loss, categorical focal loss, and categorical hinge loss, each added with a calibration component (to penalize overconfident/underconfident predictions) and a cross-entropy loss with entropy-based regularization. We demonstrate that, compared to using the de-facto cross-entropy loss function, the proposed loss functions demonstrated superior performance toward this classification task. We further improved performance by constructing prediction- and model-level ensembles. In the process, we obtained state-of-the-art performance in classifying the pediatric CXR dataset into normal, bacterial pneumonia, and viral pneumonia classes. 

Q3: However, the following revisions are required. The title of the paper needs revision.

Author response: Agreed. The title is modified as “Novel loss functions for ensemble-based medical image classification” to make it simpler and convey clarity. 

Q4: The length of the paper is very long, and I recommend authors to focus on the essential parts and discard the basic stuff such as what is ensemble deep learning, what is statistical analysis, etc.

Author response: Thanks for these insightful comments. Indeed, addressing this comment has helped improve readability in the revised manuscript. The following changes are made to the revised manuscript:

(i) Discussions regarding CXR modality-specific knowledge transfer, deep ensemble learning, and statistical analysis are removed from the introduction section for redundancy. 

(ii) Discussions regarding the existing segmentation losses, segmentation evaluation metrics, and existing classification loss functions are removed but adequate references are provided. 

(iii) The polar plots are removed for redundancy. 

Q5: Secondly, the manuscript should be focused on the loss functions for multiclass image classification. But the paper discusses so many other things, such as disease classification details. If authors want to keep these contents, then organize the manuscript so that it is easy for readers to read. I recommend, authors to revise the entire manuscript with the focus on “with loss functions for multiclass image classification.”

Author response: We sincerely thank the reviewer for these valuable comments. Addressing Q4 has helped remove redundant information and improve readability. However, this systematic study includes several steps toward attaining state-of-the-art performance in classifying the pediatric CXR data which provides an objective way of evaluating the benefits of this method. These steps include (i) performing lung segmentation to prevent learning irrelevant features in the background, (ii) training and evaluating models with the existing and proposed loss functions, (iii) improving performance by constructing prediction- and model-level ensembles, and (iv) using visualization tools to interpret the learned behavior of the models and the ensemble. We illustrate these steps in Fig. 1 and Fig. 2. We observed that the weighted averaging of the predictions of the top-3 and top-5 model-level ensembles obtained state-of-the-art performance using the pediatric CXR data. 

Q6: The literature review carried out for the proposed work is not up to date. The proposed research demands the referral of some of the latest research works published recently, such as “ReCognizing SUspect and PredictiNg ThE SpRead of Contagion Based on Mobile Phone LoCation DaTa (COUNTERACT): A System of identifying COVID-19 infectious and hazardous sites, detecting disease outbreaks based on the internet of things, edge computing, and artificial intelligence”, “Histogram of Oriented Gradient-Based Fusion of Features for Human Action Recognition in Action Video Sequences

Author response: Thanks. We have cited the COVID-19 study per the reviewer’s suggestions.

[35] Ghayvat H, Awais M, Gope P, Pandya S, Majumdar S. 2021. Recognizing suspect and predicting the spread of contagion based on mobile phone location data (counteract): a system of identifying covid-19 infectious and hazardous sites, detecting disease outbreaks based on the internet of things, edge computing, and artificial intelligence. Sustainable Cities and Society 69(12):102798

Response to Reviewer #2:

We thank the reviewer for the valuable comments on this study. 

Q1: The work lacks novelty. Literature review is poor. Proposed work should be described clearly with clear diagram/algorithm along with discussion. Introduction section and conclusion need to be revised.

Author response: The principal limitation of the de-facto cross-entropy loss is that it asserts equal learning from all the classes. This adversely impacts training and classification performance during class-imbalanced training. This holds for medical images, particularly CXRs, where a class imbalance exists between the majority normal class and other minority disease classes. Although the choice of the loss function impacts model performance, to the best of our knowledge, we observed that no literature exists that performs a comprehensive analysis and selection of an appropriate loss function toward the classification task under study. The contribution of this study includes a comprehensive statistical evaluation of several existing and proposed loss functions toward a medical image classification task. This guides the researchers regarding making an appropriate selection of a loss function for the task under study. The proposed loss functions could be applied for binary, multi-class, and multi-label classification tasks. We further improve classification performance by constructing an ensemble of models trained with the existing and proposed loss functions. In the process, we observed that the ensemble delivered superior performance compared to the individual models. 

We made sure to include relevant references from the current year. The references are formatted per PLOS ONE requirements. The citations include those published in reputed journals like IEEE, Elsevier, Springer, and MDPI.

The proposed work is briefly discussed in the introduction in lines 80 – 96. Fig. 1 and Fig. 2 illustrate the steps involved in this systematic study. 

The introduction section has been revised to remove contents for redundancy. The conclusion discusses the benefits and limitations of the current study and the scope for future study. 

Q2: I strongly suggest the authors to format the content and structure of the paper before submission. This article does not look like a research paper, just like a manual. Simultaneously, the figures included in this paper are not obvious and casual. The authors should refer to some related papers in some venue for revision.

Author response: Thanks for these insightful comments. We have made the following changes to the manuscript to improve readability. 

(i) Discussions regarding CXR modality-specific knowledge transfer, deep ensemble learning, and statistical analysis are removed from the introduction section for redundancy; (ii) Discussions regarding the existing segmentation losses, segmentation evaluation metrics, and existing classification loss functions are removed but adequate references are provided; (iii) The polar plots are removed for redundancy; (iv) We made sure to include relevant references from the current year (2021). The references are formatted per PLOS ONE requirements. The citations include those published in reputed journals from publishers such as IEEE, Elsevier, Springer, and MDPI. (v) The figures (Fig. 1 and Fig. 2) illustrate the steps involved in this systematic study. Fig. 3, Fig. 4, and Fig. 5 illustrate the performances (in terms of AUROC, AUPRC, and confusion matrix) obtained by the individual models and the prediction- and model-level ensembles. Fig. 6 illustrates Grad-CAM-based localization of the disease ROIs achieved using the trained models and the ensemble. This provides a qualitative analysis of the learned behavior by the individual models and the ensemble. 

Response to Reviewer #3:

We thank the reviewer for the appreciative and constructive comments on this study. 

Q1: Abstract should be concise yet. But should give complete overview of the work and study. Abstract should reflect the background knowledge on the problem addressed need to be added. Abstract should reflect the wide range of applications and its possible solutions need to be added.

Author response: Thanks for these insightful comments. We confirmed that the abstract does not exceed the 300 words count as recommended in the PLOS ONE submission guidelines. We modified the abstract to include the background knowledge about the problem and the proposed solution. The revised abstract is given below. Note that while we provide the link for our code, we will open the site only after the manuscript is published.

Medical images commonly exhibit multiple abnormalities. Predicting them requires multi-class classifiers whose training and desired reliable performance can be affected by a combination of factors, such as, dataset size, data source, distribution, and the loss function used to train deep neural networks. Currently, the cross-entropy loss remains the de-facto loss function for training deep learning classifiers. This loss function, however, asserts equal learning from all classes, leading to a bias toward the majority class. Although the choice of the loss function impacts model performance, to the best of our knowledge, we observed that no literature exists that performs a comprehensive analysis and selection of an appropriate loss function toward the classification task under study. In this work, we benchmark various state-of-the-art loss functions, critically analyze model performance, and propose improved loss functions for a multi-class classification task. We select a pediatric chest X-ray (CXR) dataset that includes images with no abnormality (normal), and those exhibiting manifestations consistent with bacterial and viral pneumonia. We construct prediction-level and model-level ensembles to improve classification performance. Our results show that compared to the individual models and the state-of-the-art literature, the weighted averaging of the predictions for top-3 and top-5 model-level ensembles delivered significantly superior classification performance (p < 0.05) in terms of MCC (0.9068, 95% confidence interval (0.8839, 0.9297)) metric. Finally, we performed localization studies to interpret model behavior and confirm that the individual models and ensembles learned task-specific features and highlighted disease-specific regions of interest. The code is available at https://github.com/sivaramakrishnan-rajaraman/multiloss_ensemble_models. 

Q2: In Introduction section, the drawbacks of each conventional technique should be described clearly. Introduction section can be extended to add the issues in the context of the existing work

Author response: Thanks for these comments. The drawbacks of using the de-facto cross-entropy loss function for model training and the need to propose novel loss functions are described in lines 49 – 68. The need for ensemble learning applied is discussed in lines 69 – 79. A brief overview of the proposed methodology is mentioned in lines 80 – 96 in the revised manuscript. The merits, limitations, and scope for future work are discussed in lines 429 – 459. 

Q3: What is the motivation of the proposed work?

Author response: The principal limitation of the de-facto cross-entropy loss is that it asserts equal learning from all the classes. This adversely impacts training and classification performance during class-imbalanced training. This holds for medical images, particularly CXRs, where a class imbalance exists between the majority normal class and other minority disease classes. Although the choice of the loss function impacts model performance, to the best of our knowledge, we observed that no literature exists that performs a comprehensive analysis and selection of an appropriate loss function toward the classification task under study. The contribution of this study includes a comprehensive statistical evaluation of several existing and proposed loss functions toward a medical image classification task. We further improve performance by constructing an ensemble of models trained with diverse loss functions. We observed that, unlike individual models, the weighted averaging of the predictions of top-3 and top-5 model-level ensembles delivered superior performance toward this task. This underscores that an ensemble of models trained with diverse loss functions improves performance compared to using individual models. We demonstrated these results with statistical significance analysis. 

Q4: Literature review techniques have to be strengthened by including the issues in the current system and how the author proposes to overcome the same. Research gaps, objectives of the proposed work should be clearly justified.

Author response: The issues with training the models with the de-facto cross-entropy loss function are discussed in lines 49 – 68. Considering class-imbalanced classification tasks that are common in medical images, using the cross-entropy loss and asserting equal learning to all classes would lead to a biased estimate of the performance. To overcome these limitations, the authors of [8] proposed the focal loss function that down weights the majority class and improves the learning of the minority class. Aside from the literature discussed in lines 49 – 68, the literature does not include a comprehensive study that investigates the effects of loss functions on medical image classification, particularly using CXRs. This study aims to provide a comprehensive analysis of using the existing and proposed loss functions to improve performance in a multi-class CXR classification task. We further improved performance through constructing ensembles of models trained with various loss functions. This systematic procedure is discussed in lines 80 – 96 in the revised manuscript. We observed that the models trained with the proposed loss functions delivered superior classification performance compared to the model trained on the de-facto cross-entropy loss function. The ensemble of the models trained with diverse loss functions achieved state-of-the-art performance using the pediatric CXR data used in this study. 

Q5: The writing of the paper needs a lot of improvement in terms of grammar, spellings, and presentations. The paper needs careful English polishing since there are many typos and poorly written sentences.

Author response: Thanks for these comments. We made sure to rectify the typos and grammatical errors and the revised manuscript has been proofread by a native English speaker. 

Q6: Authors can use latest related works from reputed journals like IEEE/ACM Transactions, MDPI, Elsevier, Inderscience, Springer, Taylor & Francis etc. and write the references in proper format, from year 2020-21. 

Author response: Thanks for these insightful comments. The revised manuscript includes several citations from the current year 2021. The references are formatted per PLOS ONE requirements. The citations include those published in reputed journals like IEEE, Elsevier, Springer, and MDPI. 

Q7: The authors seem to disregard or neglect some important finding in results that have been achieved in paper. So, elaborate and explain the results in more details. Improve the results and discussion section in paragraph.

Author response: Thanks for these valuable comments. We made sure to discuss the results obtained in every step of this systematic study, with statistical significance analysis. We also performed qualitative analyses to interpret the learned behavior of the trained models and the ensemble. 

Q8: The conclusion should state scope for future work.

Author response: Thanks. We have discussed the scoped for future work in lines 453 – 459 in the revised manuscript.

---

## [Decision Letter · Decision Letter 1]

1 Dec 2021

Novel loss functions for ensemble-based medical image classification

PONE-D-21-32872R1

Dear Dr. Rajaraman,

We’re pleased to inform you that your manuscript has been judged scientifically suitable for publication and will be formally accepted for publication once it meets all outstanding technical requirements.

Kind regards,

Thippa Reddy Gadekallu

Academic Editor

PLOS ONE

Additional Editor Comments (optional):

Reviewers' comments:

Reviewer's Responses to Questions

**Comments to the Author**

1. If the authors have adequately addressed your comments raised in a previous round of review and you feel that this manuscript is now acceptable for publication, you may indicate that here to bypass the “Comments to the Author” section, enter your conflict of interest statement in the “Confidential to Editor” section, and submit your "Accept" recommendation.

Reviewer #1: All comments have been addressed

Reviewer #3: All comments have been addressed

2. Is the manuscript technically sound, and do the data support the conclusions?

Reviewer #1: Yes

Reviewer #3: Yes

3. Has the statistical analysis been performed appropriately and rigorously? 

Reviewer #1: Yes

Reviewer #3: Yes

4. Have the authors made all data underlying the findings in their manuscript fully available?

Reviewer #1: Yes

Reviewer #3: Yes

5. Is the manuscript presented in an intelligible fashion and written in standard English?

Reviewer #1: Yes

Reviewer #3: Yes

6. Review Comments to the Author

Reviewer #1: Authors have addressed all the concerns. The research work should be shared with the science community.

Reviewer #3: All the comments made by the reviewers are addressed well by the authors.

No further comments are required.

7. PLOS authors have the option to publish the peer review history of their article (what does this mean?). If published, this will include your full peer review and any attached files.

Reviewer #1: **Yes: **Sharnil Pandya

Reviewer #3: No

---

## [Editor Report · Acceptance letter]

21 Dec 2021

PONE-D-21-32872R1 

Novel loss functions for ensemble-based medical image classification 

Dear Dr. Rajaraman:

I'm pleased to inform you that your manuscript has been deemed suitable for publication in PLOS ONE. Congratulations! Your manuscript is now with our production department. 

Kind regards, 

on behalf of

Dr. Thippa Reddy Gadekallu 

Academic Editor

PLOS ONE